# Research Progress of DcR3 in the Diagnosis and Treatment of Sepsis

**DOI:** 10.3390/ijms241612916

**Published:** 2023-08-18

**Authors:** Jingqian Su, Zhiyong Tong, Shun Wu, Fen Zhou, Qi Chen

**Affiliations:** Fujian Key Laboratory of Innate Immune Biology, Biomedical Research Center of South China, College of Life Science, Fujian Normal University, Fuzhou 350117, China; tongzhiyong1998@163.com (Z.T.); qsx20221412@student.fjnu.en (S.W.); 15859446856@163.com (F.Z.)

**Keywords:** decoy receptor 3, sepsis, research progress, diagnosis, treatment, biomarkers

## Abstract

Decoy receptor 3 (DcR3), a soluble glycosylated protein in the tumor necrosis factor receptor superfamily, plays a role in tumor and inflammatory diseases. Sepsis is a life-threatening organ dysfunction caused by the dysregulation of the response to infection. Currently, no specific drug that can alleviate or even cure sepsis in a comprehensive and multi-level manner has been found. DcR3 is closely related to sepsis and considerably upregulated in the serum of those patients, and its upregulation is positively correlated with the severity of sepsis and can be a potential biomarker for diagnosis. DcR3 alone or in combination with other markers has shown promising results in the early diagnosis of sepsis. Furthermore, DcR3 is a multipotent immunomodulator that can bind FasL, LIGHT, and TL1A through decoy action, and block downstream apoptosis and inflammatory signaling. It also regulates T-cell and macrophage differentiation and modulates immune status through non-decoy action; therefore, DcR3 could be a potential drug for the treatment of sepsis. The application of DcR3 in the treatment of a mouse model of sepsis also achieved good efficacy. Here, we introduce and discuss the progress in, and suggest novel ideas for, research regarding DcR3 in the diagnosis and treatment of sepsis.

## 1. Introduction

Decoy receptor 3 (DcR3) is a soluble glycosylated protein, primarily circulating as a secreted protein and is highly expressed in tumors and inflammatory diseases [1]. DcR3 can competitively bind FasL (TNF superfamily receptor 6, TNFSF6), LIGHT (tumor necrosis factor ligand superfamily member 14, TNFSF14), and TL1A (TNF-like molecule 1A) with a strong affinity through decoy action and can block downstream signaling. Furthermore, it can regulate macrophage and lymphocyte differentiation through non-decoy action and is a multipotent immunomodulator [2,3]. Sepsis is a critical condition characterized by the malfunctioning of organs due to the dysregulation of a patient’s immune response and is triggered by infection. It has serious consequences, such as high morbidity, mortality, treatment costs, and poor prognosis, and places a great burden on global healthcare systems [4,5,6]. It is estimated that the U.S. healthcare system spends at least $20 billion annually on sepsis care. The harm caused by sepsis will likely increase as the human population ages [7].

DcR3 is closely related to sepsis. Significant upregulation of DcR3 levels in the serum of patients with sepsis is positively correlated with its severity and could therefore be used as a biomarker for its diagnosis [8]. DcR3 alone or in combination with other markers has shown promising results in the early diagnosis of sepsis, due to the multiple functions of DcR3 in immunomodulation. Previously, DongYu Liang et al. (2015) found that in vitro injection of DcR3–Fc (decoy receptor 3 fused to an Fc fragment of IgG1) recombinant protein increased survival, decreased inflammation levels, and improved organ damage in septic mice [9]. In this review, we aim to discuss the existing research on the potential for the involvement of DcR3 with sepsis, deepen our understanding of DcR3, and provide new ideas for the diagnosis and treatment of sepsis.

## 2. DcR3

DcR3, alternatively referred to as tumor necrosis factor receptor superfamily member 6b (TNFRSF6B), TR6, or M68, belongs to the decoy receptor family within the tumor necrosis factor receptor superfamily. It is a soluble receptor that lacks a structural transmembrane domain and is chemically a glycosylated protein [10]. As a member of the decoy receptor family, members of the DcR3 family also include decoy receptor 1 (DcR1), decoy receptor 2 (DcR2), and osteoprotegerin (OPG), of which DcR3 has the highest homology to OPG, with both being secreted proteins [11]. The DcR3 gene was first identified and named by Pitti et al. in 1998, and the DcR3 protein was highly expressed in the serum of patients with malignancies [12,13]. The DcR3 gene is present in several species, except for rats (*Rattus norvegicus*) and mice (*Mus musculus*). The human (*Homo sapiens*) DcR3 gene is located at the telomeric end of chromosome 20 (20q13.3) and encodes 300 amino acids, of which the first 29 amino acid residues are signal peptide sequences with a mature product size of approximately 33 kilodalton (kDa) [14].

DcR3 is not expressed or is less expressed under normal physiological conditions, but is highly expressed in patients with gastric [15] or pancreatic cancer [16], and patients with burns [17], autoimmune diseases [18], sepsis [8], or chronic obstructive pulmonary disease [19]. After bacterial or fungal infection, invading pathogens are captured, processed, and presented by antigen-presenting cells (APCs), such as macrophages and dendritic cells (DCs). As shown in Figure 1, pathogen-associated molecular patterns (PAMPs) such as lipopolysaccharide (LPS), phospholipid wall acid (LTA), and yeast glycan (zymosan) can bind to toll-like receptor 2/4 (TLR2/4) on the cell membrane, thus activating downstream signaling pathways such as nuclear factor kappaB (NF-κB) and phosphoinositide 3 kinase-protein kinase B (PI3K-AKT), promoting DcR3 expression-related gene transcription and thus leading to an increased DcR3 content [17,20,21,22,23].

The physiological role of DcR3 is not fully understood, but as a soluble receptor in the decoy receptor family, DcR3 can neutralize the biological effects of FasL, LIGHT, and TL1A through its decoy function [24]. As shown in Figure 2, DcR3 can competitively bind these ligands due to higher affinity, blocking downstream signaling and exerting anti-apoptotic and anti-inflammatory effects by reducing the original biological activity of the ligand [12,25,26,27]. Furthermore, DcR3 exerts a non-decoy function through acetyl heparan sulfate proteoglycan (HSPG), which can regulate macrophage differentiation toward the M2 type, regulate the T-helper 1/2 cells (Th1/Th2) bias toward Th2 differentiation, and promote monocyte cell adhesion [28].

FasL, TL1A, and LIGHT can activate downstream signaling pathways by binding to specific receptors on the cell membrane, leading to cellular inflammation and apoptosis. FasL is predominantly synthesized by activated T-lymphocytes, specifically cytotoxic T-lymphocytes (CTL), as well as natural killer (NK) cells. The binding of FasL to its receptor Fas (CD95) triggers apoptosis in target cells [29,30]. The FasL–Fas system has been established as a central regulator of apoptosis in mammalian cells [31]. Furthermore, the FasL–Fas system can induce inflammatory responses in various cell types, induce the expression of pro-inflammatory cytokines such as interleukin-6 (IL-6) and tumor necrosis factor alpha (TNF-α), and recruit neutrophils to sites of inflammation [32].

LIGHT is expressed in lymphoid tissues and can bind to three receptors: herpesvirus entry mediator (HVEM), lymphotoxin β receptor (LTβR), and DcR3 [33,34]. The primary effects of LIGHT–LTβR activation include induction of apoptosis and, in most cases, the LIGHT–HVEM interaction also selectively activates NF-κB, which initiates inflammation-associated gene transcription [35].

TL1A was originally found in endothelial cells and is a ligand for death receptor 3 (DR3) [36]. Activation of TL1A–DR3 can induce apoptosis and activate multiple signaling pathways (NF-κB, MAPK, etc.), thus inducing the production of inflammatory factors [37,38]. Furthermore, by binding to DR3, endogenous TL1A inhibits endothelial cell proliferation and angiogenesis in an autocrine or paracrine manner and exerts tumor-suppressive effects [39,40].

Besides exercising a decoy function, DcR3 can also perform a non-decoy function, i.e., DcR3 acts as an effector molecule that directly regulates the activity of many cell types [41]. Soluble DcR3–Fc binds to CD14+ monocytes and regulates their differentiation and maturation into DCs; CD4+ T cells co-cultured with DcR3–Fc-treated DCs proliferate in the Th2 phenotype; DcR3 drives the conversion of macrophages to the M2 phenotype; and DcR3 both induces actin reorganization and increases cell adhesion in human monocytes [28,42,43,44,45,46].

In recent years, DcR3 has been reported to be closely associated with HSPG, and through it may exert a non-deceptive function [47]. The DcR3 protein structure contains a segment of the heparin sulfate proteoglycan binding domain (HBD), which contains basic amino acid residues (aa 256–261) that activate HSPG. Syndecan-2 (SDC2) and CD44v3 are the main HSPGs in immune cells, DcR3 binds to Syndecan-2 and CD44v3, and the interaction between DcR3 and HSPGs is attenuated by acetyl heparan sulfate (HS) [47,48,49]. HBD–Fc cannot bind to FasL, LIGHT, and TL1A, but both HBD–Fc and DcR3–Fc can induce macrophages to differentiate to the M2 type, suggesting that the non-deceptive effect of DcR3 may be achieved by binding to HSPG [14,44,50].

Based on the decoy and non-decoy functions, DcR3 can maintain the homeostasis of the internal environment in inflammatory diseases by reducing the apoptosis of immune cells and decreasing the level of inflammatory factors. Therefore, the administration of DcR3 by increasing its autologous expression or its expression in vitro could be a research direction in the treatment of inflammatory diseases [51].

## 3. Sepsis

Sepsis is a critical condition characterized by the dysfunction of vital organs, resulting from an infection-triggered dysregulation of the patient’s immune response. No specific drug has yet been found that can alleviate or even cure sepsis in a comprehensive and multi-level manner [52]. Sepsis is a common disease, with approximately half a million cases occurring worldwide each year. In 2017, 11 million sepsis-related deaths were reported, representing a mortality rate of 19.7% of total global deaths. Furthermore, sepsis morbidity and mortality show large regional differences, with the burden being the highest in less developed regions represented by sub-Saharan Africa [53,54]. With an increase in the ageing population and with its various complications, sepsis will place a greater economic burden on the world [55]. As a global disease with high morbidity, and mortality, sepsis was listed as a global health priority by the World Health Organization (WHO) in 2017 [56,57,58].

The word sepsis is derived from the Greek word *σήψη*, meaning putrefaction. The term has been used since the time of Hippocrates, but the definition has been ambiguous in the medical community, leading to numerous difficulties in clinical diagnosis and medication prescription [56,59]. To overcome this inconsistency, the definition of sepsis has been continuously revised in recent decades. The original definition of sepsis (sepsis 1.0) was developed at the consensus meeting of the American College of Chest Physicians/Society of Critical Care Medicine in 1991, subsequently updated in 2001 (sepsis 2.0), and the most recent definition in use today is sepsis 3.0, published in 2016 [5,60,61]. Sepsis 3.0 is defined as a life-threatening organ dysfunction caused by the dysregulation of the body’s response to infection, and septic shock is the most severe manifestation of sepsis with severe circulatory, cellular, and metabolic dysfunction and a greater risk of death [5,62]. Sepsis 3.0 emphasizes that sepsis is caused by infection and can be distinguished from non-infectious causes of organ dysfunction and common infections that do not cause organ dysfunction.

The pathophysiological mechanisms of sepsis are complex—it is a syndrome with high heterogeneity. This is reflected primarily in the different clinical manifestations, as patients of different ages exhibit different physiological characteristics, and this heterogeneity is considered a major factor in the failure of immunomodulatory tests in patients with sepsis. Hyperinflammation and immunosuppression are intertwined in the course of the disease. As shown in Figure 3, pathogenic infections such as bacteria, fungi, viruses, and parasites cause systemic inflammation in the body, and, under normal conditions, the pro-inflammatory and anti-inflammatory systems in the body are balanced. However, once the balance is disrupted, and if it is not restored in time, excessive inflammation or immunosuppression may occur, which in turn can lead to the development of sepsis. Subsequently, a series of irreversible damages can occur, such as mitochondrial damage, thrombosis, and apoptosis, resulting in a huge disease burden [63,64].

According to the definition offered by sepsis 3.0, sepsis is a disease caused by an overreaction of the host to infection. The occurrence of sepsis originates from various infections, including pathogenic microbial infections and various traumas, the former being more common in clinical practice. As shown in Figure 4, after an exogenous pathogen invades the organism, pattern recognition receptors (PRRs) recognize and bind PAMPs and use them to activate the innate immune system, while the complement system is also activated. Complement component 3a (C3a) and complement component 5a (C5a) are then produced and released in large quantities, leading to massive production and release of pro-inflammatory cytokines, resulting in an inflammatory storm. Elevated inflammatory conditions induce cellular harm and prompt the liberation of damage-related molecular patterns (DAMPs), including high mobility group protein 1 (HMGB1), thereby intensifying the generation and discharge of inflammatory mediators [65,66,67]. The hyperinflammatory environment in the body also promotes the activation of the coagulation system and intravascular microthrombosis, leading to disorders of the cardiovascular system. Prolonged excessive inflammation disrupts immune homeostasis and forces the body to compensate for the production of large amounts of anti-inflammatory molecules. Without timely intervention, excessive amounts of inflammatory molecules can cause immunosuppression, resulting in massive apoptosis of T-cells and immune deficiency [68]. The immunity of the organism is not sufficient to fight secondary infections, often leading to death.

The damage caused by sepsis to the organism worsens with delayed intervention and therefore requires early recognition and standardized treatment [69]. Mortality in patients with sepsis increases by 5–10% for every 1 h delay in treatment [70]. Early diagnosis is a prerequisite for improving the recovery rate and can greatly reduce the morbidity and mortality of sepsis. Blood cultures remain the gold standard for diagnosis as this method allows the isolation and identification of pathogens and further tests for antimicrobial susceptibility. However, these are time-consuming, prone to false-negative results and cannot be used to diagnose non-bacterial causes of sepsis, which is conducive neither to the early diagnosis of sepsis nor to the timely intervention of antibiotics [71]. Biomarkers play an important role in the diagnosis, treatment, and prognosis of diseases. As shown in Table 1, sepsis biomarkers include acute phase response proteins, cytokines and chemokines, cell surface and soluble receptors, vascular endothelium-related markers, coagulation-related markers, neurotransmitter-related markers, and hormone-related markers [72,73,74,75,76,77]. Accurate diagnosis of sepsis is difficult to achieve with any single biomarker, and the combination of multiple biomarkers and sepsis scores can improve the accuracy of sepsis diagnosis and prognosis determination [78].

Severity and prognosis evaluation can help design an early and effective clinical treatment plan [79]. Sepsis scoring tools are diverse and continuously evolving as the definition of sepsis is updated. The systemic inflammatory response syndrome (SIRS) score, or sepsis 1.0 diagnostic criteria, was proposed in 1991 and has been widely used since its development. Despite the development of diagnostic criteria for sepsis 2.0 in 2001, the SIRS score was used until 2016 as the criteria were too vague; however, SIRS is too sensitive and is prone to misclassifying most patients with mild inflammation as also being septic, resulting in increased morbidity, decreased mortality, and a waste of healthcare resources [80,81]. In 2016, sepsis 3.0 diagnostic criteria were proposed and the Sequential Organ Failure Assessment (SOFA) scoring system was recommended. The SOFA score has better predictive power than the SIRS score and more closely resembles the characteristics of the onset of sepsis [57]. However, because the SOFA score requires the testing of six organ systems, the waiting time is long. For the early identification of sepsis, the Rapid Sequential Organ Failure (qSOFA) score was also proposed, which is quick and easy, reproducible, does not require complete medical facilities, and is therefore useful in areas with scarce medical resources [82]. Furthermore, various scoring tools have been developed such as lactate ≥2.0 mmol/L combined with the qSOFA (LqSOFA) score, the British National Early Warning Score (NEWS), and the Acute Physiological and Chronic Health Status (APACHE II) score, which are intended to detect and treat sepsis in the shortest possible time, thus improving patient survival and reducing disease burden [83,84,85].

Several drugs have been developed to treat sepsis, but no satisfactory results have yet been achieved. Only early prevention, detection, and treatment can reduce the incidence and improve the survival rate of patients with sepsis. Treatment options for sepsis have focused on fluid resuscitation, antibiotic therapy, and immunotherapy [86,87]. Although the short-term morbidity and mortality of sepsis can be considerably reduced through this treatment, the long-term morbidity and mortality rate remain high. Fluid resuscitation is one of the primary therapeutic measures for sepsis. In recent years, updates to sepsis fluid resuscitation treatment protocols have focused on various aspects, such as treatment initiation timing, fluid selection, fluid volume, and monitoring indicators. The “Save the Sepsis Campaign” guidelines recommend starting an intravenous infusion of one or more broad-spectrum antibiotics within 1 h of the diagnosis of sepsis [88,89]. The ever-shortening window of antibiotic use has forced medical personnel to sacrifice diagnostic accuracy in pursuit of speed, inevitably leading to the overuse of antibiotics [90].

The aim of immunotherapy is to restore or strengthen the body’s immunity. Abnormalities such as dysregulation of cytokine secretion, decreased APC function, and apoptosis and depletion of lymphocytes occur during sepsis. Immunotherapeutic agents targeting sepsis include cytokine classes, such as interferon γ (IFN-γ) and thymidine α1, and cell surface co-inhibitory molecules, such as anti-programmed cell death receptor 1 (PD-1) antibodies, and anti-programmed cell death receptor ligand 1 (PD-L1) antibodies [68]. Anti-PD-1 antibodies, PD-L1 antibodies, and interferon γ have shown good promise in animal models [91]. Furthermore, vitamin C, vasopressin, and Chinese herbal medicine play an active role in sepsis treatment [92,93].

## 4. DcR3 as a Biomarker for Sepsis

Early diagnosis and prompt treatment of patients with sepsis can improve the survival rate and reduce the abuse of antibiotics, thereby postponing the emergence of drug-resistant bacteria. Blood culture is the gold standard for sepsis detection, but this test is time consuming (48–72 h), has a low positive rate and an inability to diagnose sepsis caused by non-bacterial infections, therefore, there is an urgent need to find some efficient and sensitive testing methods for use in clinics [94]. The discovery of specific biomarkers has alleviated this situation, with early diagnosis of sepsis relying on specific biomarkers, but these also have certain limitations [78]. From a clinical perspective, the preference is to find a single and highly specific marker, however, evidence-based medicine has shown that the application of single biomarkers has limitations, such as a lack of predictive accuracy and tissue specificity; a combination protocol is therefore recommended. This also has limitations such as there being few combinations, poor combination protocols, and high testing costs. As a result, the only way to predict sepsis progression accurately is to continuously expand the range of biomarkers and to use multifaceted and multiple combinations of protocols.

DcR3 is highly expressed in inflammatory diseases, but the exact role of highly expressed DcR3 in the inflammatory process remains to be elucidated [15,95,96]. Clinical data have revealed that the average plasma DcR3 level was 0.17 ng/mL in normal individuals and 4.25 ng/mL in patients with sepsis, which was 25-fold higher compared with normal patients. Furthermore, the higher the APACHE II score, the higher the DcR3 level in patients with sepsis, suggesting that the DcR3 level is positively correlated with the severity of sepsis in patients [8]. Moreover, DcR3 is universally present in different species and, therefore, can be used as a sepsis biomarker [8,28,97,98]. DcR3 can also differentiate sepsis from SIRS, a systemic inflammatory disease caused by non-infectious factors that is primarily treated with steroids. As non-infectious factors do not cause changes in DcR3 levels, the use of DcR3 as a biomarker for sepsis can help reduce the use of steroids and provide timely antibiotic therapy. Owing to the characteristics of the DcR3-inducible expression mechanism, viral infections do not cause elevated DcR3 levels; therefore, DcR3 can help differentiate bacterial and viral infections. Furthermore, DcR3 levels can indicate if the treatment regimen is effective and working, guiding the effective clinical use of drugs. An increase in DcR3 levels can indicate that the treatment regimen is ineffective, whereas a decrease in DcR3 can indicate improvement and recovery.

Although DcR3 is an emerging marker of sepsis, it is rarely used alone and, therefore, the use of DcR3 combined with other substances has become an emerging trend in the prediction of sepsis. The combination of serum DcR3 levels and the neutrophil-to-lymphocyte ratios in the blood can effectively assess the prognosis of patients with sepsis. The incorporation of the triple combination, comprising DcR3, the soluble urokinase-type fibrinogen activator receptor, and procalcitonin (PCT), enhances the accuracy and precision of sepsis detection [98].

## 5. DcR3 as a Drug for Sepsis Treatment

With an enhanced understanding of the pathogenesis of sepsis, immunotherapeutic agents are receiving increasing attention. The progression of sepsis can be broadly categorized into two distinct phases: hyperinflammation in the early stages, and immunosuppression in the late stages [99,100,101]. Hyperinflammation is caused by the excessive activation of the innate immune system of the body, leading to clinical symptoms, such as hyperthermia, hypotension, and shortness of breath, which can be life-threatening. Abnormal coagulation plays a key role in hyperinflammation, and hypovascular endothelial cell function is a prominent hallmark of sepsis [102,103]. DcR3 regulates body temperature, reduces inflammatory and chemokine levels, regulates the function of vascular endothelial cells, and can protect human umbilical vein endothelial cells from LPS-induced apoptosis [9,104]. Immunosuppression emerges as the primary etiological factor contributing to mortality in individuals afflicted with sepsis [105]. The immunosuppressive phase is marked by a drastic decrease in lymphocytes, leading to a low immune function and consequent inability to defend the patient against invading pathogenic microorganisms. As shown in Figure 5, DcR3 can suppress inflammation and promote the differentiation of macrophages into M2 type and T-cells into Th2 type.

Endogenous DcR3 is likely unable to improve sepsis as it is induced in low amounts and cannot establish an overwhelming advantage. The in vitro expressed DcR3–Fc recombinant protein was injected intravenously into a mouse model of cecum ligation and puncture-induced sepsis, and significantly improved survival, decreased inflammatory factor levels, improved organ lesions, inhibited lymphocyte apoptosis, and increased lymphocyte numbers were observed [9]. The recombinant protein also reduced the secretion of pro-inflammatory cytokines and chemokines induced by the influenza A virus, thereby attenuating lung infiltration and reducing lethality [104]. DcR3 transgenic mice had reduced levels of pro-inflammatory cytokines and chemokines in bronchoalveolar lavage fluid, and decreased lung infiltration [104], suggesting that the development of DcR3 for sepsis treatment is promising.

## 6. Summary and Outlook

Although thousands of experimental animals have been rescued from sepsis in several preclinical studies over the past decades, no specific drug has yet emerged that can significantly improve the prognosis of human patients [106]. DcR3 is expressed by the human body under pathological conditions, is derived from itself, and is compatible with the human body. DcR3 can inhibit lymphocyte apoptosis and alleviate the body’s immunosuppression to a certain extent by playing a decoy role. DcR3 can also regulate T-cell and macrophage differentiation and alleviate excessive inflammation. Therefore, the use of DcR3 is a promising drug in the treatment of sepsis. Animal experiments have confirmed that the use of DcR3 for sepsis treatment is feasible and effective [8].

Future research on DcR3 should focus on elaborating the mechanism of action of DcR3 in ameliorating sepsis, as well as promoting the clinical application of DcR3. Because the DcR3 gene does not exist in the rat and mouse genome, it is not possible to directly construct DcR3 knockout mice. Additionally, although the therapeutic effects of DcR3 in septic mice can be investigated by constructing DcR3-expressing mice or by administering DcR3 in vitro, this is a more demanding experimental technique for the operator and is both time-consuming and expensive, making the application of DcR3 in the murine model somewhat limited. DcR3–Fc is a recombinant fusion protein formed by linking the DcR3 sequence to a human IgG1 Fc fragment, and DcR3–Fc can better mimic the biological effects of DcR3. The intrathecal administration of recombinant DcR3–Fc fusion proteins demonstrates a potential therapeutic effect in experimental autoimmune encephalomyelitis, suppresses macrophage activation induced by influenza virus, and mitigates pulmonary inflammation and mortality. Consequently, DcR3–Fc exhibits promise as a valuable tool for further investigation and facilitates comprehensive studies on DcR3 [107].

DcR3 can be used as a biomarker for sepsis and can be used alone for early diagnosis or in combination with markers such as C-reactive protein (CRP), PCT, butyrylcholinesterase, presepsin, and pro-adrenomedullin. Although good results have been achieved by combining DcR3 with other sepsis markers, its application as a biomarker is partially limited because gram-positive, gram-negative, and fungal infections can all cause the upregulation of serum DcR3 levels in patients, making it impossible to effectively and specifically distinguish the type of infectious agent. The effective combination of DcR3 with other markers still needs to be explored further.

The two stages of the disease in patients with sepsis—excessive inflammation and immunosuppression—are dynamically transformed and both can exist simultaneously [99]; therefore, drugs that are effective in treating both are required. However, these two phases have opposite characteristics. Excessive inflammation is caused by an overly strong inflammatory response of the body; therefore, the therapeutic drugs in this phase need to reduce the body’s immunity, thus reducing the level of inflammation in the body. The immunosuppression phase is caused by excessive death of the body’s lymphocytes, resulting in low immunity, and the therapeutic drugs in this phase need to save lymphocytes from death and activate the body’s immune function. Therefore, to regulate sepsis, it is necessary to exert different pharmacological effects by modulating the dosage to reduce inflammation, while rescuing lymphocytes from apoptosis. This is the reason for the lack of effective drugs for sepsis treatment on the market. Although DcR3 can reduce the level of inflammation through non-decoy action and rescue lymphocytes from apoptosis through the competitive binding of apoptosis-related ligands through decoy action, the specific mechanism, efficacy, and safety of its action still needs further study.

Sepsis is a highly heterogeneous syndrome, and the diversity of patients at the individual level, including different clinical presentations and physiological characteristics at different ages, will affect the diagnosis, treatment standardization protocols, implementation, and development of sepsis [108]. With the rapid development of modern technology, big data and artificial intelligence will play an important role in the prevention, diagnosis, and treatment of sepsis [109,110]. Genetics and other factors play important roles in the pathogenesis of sepsis and its outcome, and this complex scenario will drive the continuous adjustment of classification methods. In the future, patients will be categorized according to their most basic biological characteristics and treated with protocols that better match their real pathology, for individualized precision medicine [61]. It is believed that in the near future, the relationship between DcR3 and sepsis and the mechanism of DcR3 treatment of sepsis will be further elucidated.

## Figures and Tables

**Figure 1 ijms-24-12916-f001:**
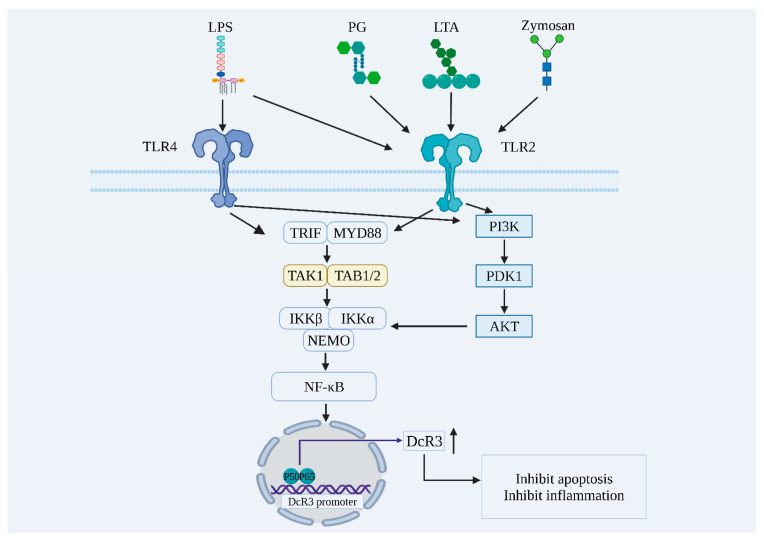
Exogenous pathogens promote DcR3 expression through the activation of NF-κB, PI3K-AKT, and other signaling pathways. LPS: lipopolysaccharide; PG: peptidoglycan; LTA: lipoteichoic acid; TLR4: toll-like receptor 4; TLR2: toll-like receptor 2; TRIF: tir domain-containing adaptor inducing interferon-beta; MYD88: myeloid differentiation factor 88; TAK1: transforming growth factor beta-activated kinase 1; TAB1/2: transforming growth factor beta-activated kinase 1 binding protein 1/2; IKKβ: IkappaB kinase β; IKKα: IkappaB kinase α; NEMO: nuclear factor kappaB essential modulator; PI3K: phosphoinositide 3 kinase; PDK1: 3-phosphoinositide-dependent protein kinase 1; AKT: protein kinase B; NF-κB: nuclear factor kappaB; DcR3: decoy receptor 3; P50: nuclear factor kappaB1; P65: nuclear factor kappa B2.

**Figure 2 ijms-24-12916-f002:**
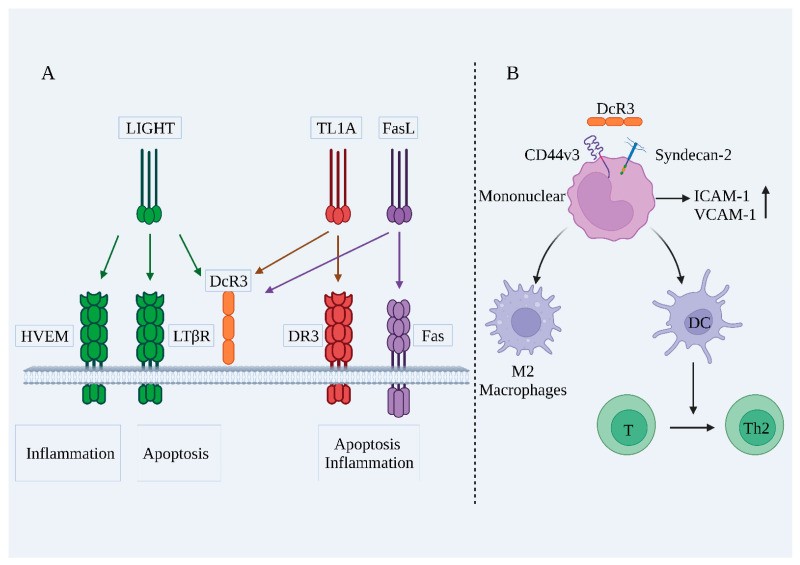
Biological functions of DcR3. (**A**) The decoy function of DcR3; (**B**) The non-decoy function of DcR3. LIGHT: tumor necrosis factor ligand superfamily member 14; TL1A: TNF-like molecule 1A; FasL: fas ligand; HVEM: herpesvirus entry mediator; LTβR: lymphotoxin beta receptor; DcR3: decoy receptor 3; DR3: death receptor 3; Fas: TNF superfamily receptor 6; CD44v3: cluster of differentiation 44 variant 3; ICAM-1: intercellular adhesion molecule 1; VCAM-1: vascular cell adhesion molecule-1; DC: dendritic cell; T: T lymphocytes; Th2: T-helper 2 cells.

**Figure 3 ijms-24-12916-f003:**
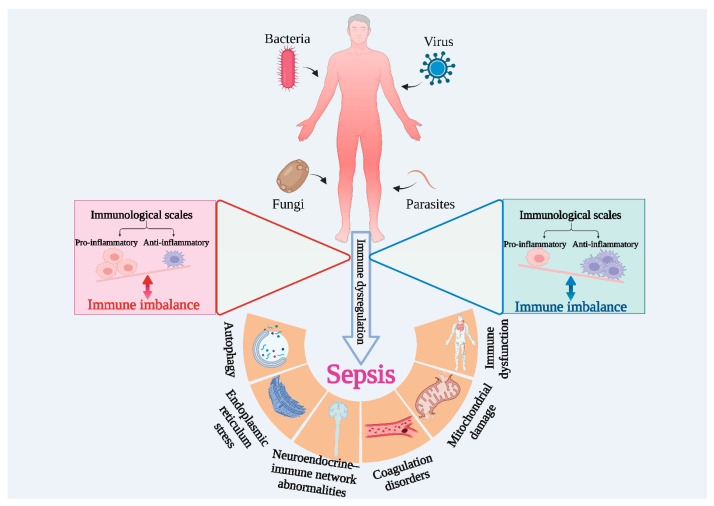
Immune imbalance is the main feature of sepsis.

**Figure 4 ijms-24-12916-f004:**
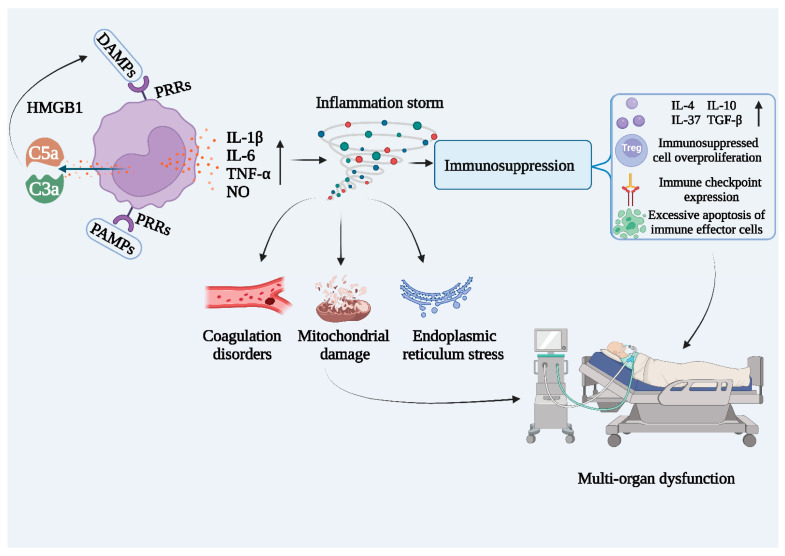
The developmental process of sepsis. DAMPs: damage-associated molecular patterns; PAMPs: pathogen-associated molecular patterns; PRRs: pattern recognition receptors; HMGB1: high mobility group box-1 protein; C3a: complement component 3a; C5a: complement component 5a; IL-1β: interleukin-1beta; IL-6: interleukin-6; TNF-α: tumor necrosis factor alpha; NO: nitrous oxide; IL-4: interleukin-4; IL-10: interleukin-10; IL-37: interleukin-37; TGF-β: transforming growth factor-beta; Treg: regulatory T-cells.

**Figure 5 ijms-24-12916-f005:**
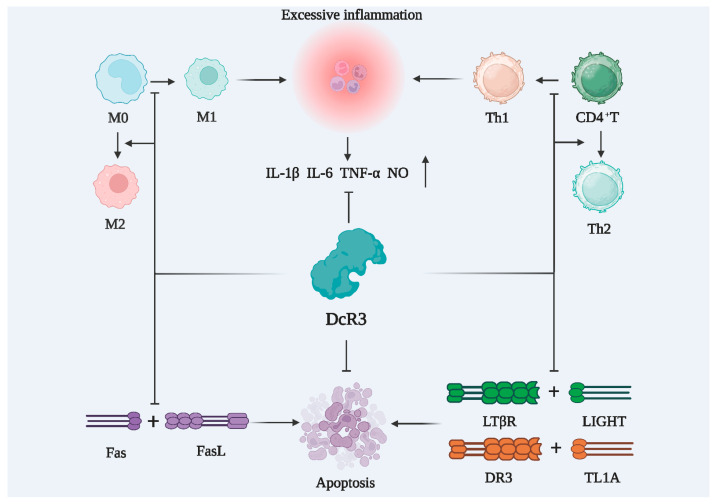
DcR3 inhibits cellular inflammation and reduces the apoptosis of immune cells. M0: M0-like macrophage; M1: M1-like macrophage; M2: M2-like macrophage; CD4+T: CD4 positive lymphocytes; Th1: T-helper 1 cells; Th2: T-helper 2 cells; IL-1β: interleukin-1beta; IL-6: interleukin-6; TNF-α: tumor necrosis factor alpha; NO: nitrous oxide; DcR3: decoy receptor 3; Fas: TNF superfamily receptor 6; FasL: fas ligand; LIGHT: tumor necrosis factor ligand superfamily member 14; LTβR: lymphotoxin-beta receptor; TL1A: TNF-like molecule 1A; DR3: death receptor 3.

**Table 1 ijms-24-12916-t001:** Relevant biomarkers of sepsis.

Function	Markers
Acute phase reactive protein	PCT; CRP
Cytokines and chemokines	IL-6; IL-10
Cell surface and soluble receptors	HLA-DR; CD64
Vascular endothelium-related	cell adhesion molecules; angiopoietin
Coagulation-related	antithrombin III
Neurotransmitter-related	butyrylcholinesterase
Hormone-related	presepsin; pro-adrenomedullin

PCT: procalcitonin; CRP: C-reactive protein; IL-6: interleukin-6; IL-10: interleukin-4; HLA-DR: human leukocyte antigen DR; CD64: cluster of differentiation 64.

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
