# Peer review of "Research Progress of DcR3 in the Diagnosis and Treatment of Sepsis"

_ijms, 2023, doi:10.3390/ijms241612916_

Round 1
Reviewer 1 Report
I would like to congratulate the authors on their fascinating work regarding this interesting article on the Progress of DcR3 in the Diagnosis and Treatment of Sepsis. The manuscript is well-written and the incorporated tables and figures make the study easy to follow.
I strongly recommend acceptance for publication of the paper after major revision.
1) I would like a brief discussion on the role of Butyrylcholinesterase levels as a Biomarker for Sepsis
consider citing:
https://www.ejso.com/article/S0748-7983(22)00811-3/fulltext
https://journals.sagepub.com/doi/10.1177/0885066616636549
2) "Sepsis is a serious clinical syndrome that is associated with high morbidity and mortality rates for patients suffering from it. It is nowadays considered a major public health problem, as it is estimated that it costs more than $20 billion annually for US health care systems"
Add this information to the introduction section and consider citing:
https://pubmed.ncbi.nlm.nih.gov/35371356/
Author Response
August 10, 2023
Prof. Dr. Maurizio Battino
Editor-in-Chief
Ms. Beiner Zhang
Assistant Editor
International Journal of Molecular Sciences
Dear Editor,
I wish to re-submit the manuscript titled “Research Progress of DcR3 in the Diagnosis and Treatment of Sepsis.” The manuscript ID is ijms-2505432.
We express our gratitude to you and the reviewers for your valuable suggestions and insightful perspectives. The manuscript has greatly benefited from these astute recommendations.
Attached is the revised version of our manuscript. In the following pages are our point-by-point responses to each of the comments of the reviewers. Revisions in the text are highlighted by the utilization of the color red. We hope that the revisions in the manuscript and our accompanying responses would be sufficient to make our manuscript suitable for publication in International Journal of Molecular Sciences.
Thank you for your consideration. I look forward to hearing from you.
Sincerely,
Jingqian Su, Ph.D.
Associate Professor
Fujian Key Laboratory of Innate Immune Biology
Biomedical Research Center of South China
College of Life Science, Fujian Normal University
Fuzhou 350117, Fujian, China
Tel: +86-18950498937
E-mail: sjq027@fjnu.edu.cn
Responses to the comments of Reviewer #1
1) I would like a brief discussion on the role of Butyrylcholinesterase levels as a Biomarker for Sepsis. consider citing:
https://www.ejso.com/article/S0748-7983(22)00811-3/fulltext
https://journals.sagepub.com/doi/10.1177/0885066616636549
Response:
Many thanks to the editors for their valuable inputs to improve the article. As reviewer mentioned, butyrylcholinesterase is one of many sepsis biomarkers. A significant decrease in serum butyrylcholinesterase levels occurs when a patient develops sepsis, and a decrease in serum butyrylcholinesterase levels can be used as a predictor of sepsis onset.
In this article, there was already a section devoted to sepsis-related biomarkers (Table 1), so butyrylcholinesterase has been placed in Table 1 and categorized as a neurotransmitter-related marker based on the biological role of butyrylcholinesterase. Both of your suggested references have been cited as [74] and [75]. It was also noted in the Discussion and Outlook that DcR3 could be considered in the future for sepsis prediction in combination with butyrylcholinesterase and others. The specific modifications are described below:
(1) Lines 217-220
As shown in Table 1, sepsis biomarkers include acute phase response proteins, cytokines and chemokines, cell surface and soluble receptors, vascular endothelium-related markers, coagulation-related markers, neurotransmitter-related markers, and hormone-related markers [72-77].
(2) Line 224
Table 1. Relevant biomarkers of sepsis.
Function |
Markers |
Acute phase reactive protein |
PCT; CRP |
Cytokines and chemokines |
IL-6; IL-10 |
Cell surface and soluble receptors |
HLA-DR; CD64 |
Vascular endothelium-related |
cell adhesion molecules; angiopoietin |
Coagulation-related |
antithrombin III |
Neurotransmitter-related |
butyrylcholinesterase |
Hormone-related |
presepsin; pro-adrenomedullin |
(3) Lines 371-373
DcR3 can be used as a biomarker for sepsis, and can be used alone for early diagnosis or in combination with markers such as CRP, PCT, butyrylcholinesterase, presepsin, and pro-adrenomedullin.
(4) Lines 607-613
- Mulita, F.; Verras, G.-I.; Bouchagier, K.; Dafnomili, V.-D.; Perdikaris, I.; Perdikaris, P.; Samaras, A.; Antzoulas, A.; Iliopoulos, F.; Akinosoglou, K.; Velissaris, D.; Bousis, D.; Maroulis, I., Butyrylcholinesterase levels as a predictive factor of septic complications development in the postoperative period of colorectal patients: Univariate analysis and predictive modeling. Eur. J. Surg. Oncol. 2023, 49 (1), e15. DOI: https://doi.org/10.1016/j.ejso.2022.11.083
- Bahloul, M.; Baccouch, N.; Chtara, K.; Turki, M.; Turki, O.; Hamida, C. B.; Chelly, H.; Ayedi, F.; Chaari, A.; Bouaziz, M., Value of Serum Cholinesterase Activity in the Diagnosis of Septic Shock Due to Bacterial Infections. J. Intensive. Care. Med. 2017, 32 (5), 346-352. DOI: 10.1177/0885066616636549
2) "Sepsis is a serious clinical syndrome that is associated with high morbidity and mortality rates for patients suffering from it. It is nowadays considered a major public health problem, as it is estimated that it costs more than $20 billion annually for US health care systems"
Add this information to the introduction section and consider citing:
https://pubmed.ncbi.nlm.nih.gov/35371356/
Response:
We would like to express our profound appreciation to the reviewer for the invaluable suggestions. The addition of relevant statistics completes the article, and the reference number is [7] (Lines 36-38 and Lines 435-437), the details are as follows:
(1) Lines 36-38
It is estimated that the U.S. healthcare system spends at least $20 billion annually on sepsis care. The harm caused by sepsis will likely increase as the human population ages [7].
(2) Lines 435-437
- Mulita, F.; Liolis, E.; Akinosoglou, K.; Tchabashvili, L.; Maroulis, I.; Kaplanis, C.; Vailas, M.; Panos, G., Postoperative sepsis after colorectal surgery: a prospective single-center observational study and review of the literature. Prz. Gastroenterol. 2022, 17 (1), 47-51. DOI: 10.5114/pg.2021.106083

Reviewer 2 Report
In this systemic review, " Research Progress of DcR3 in the Diagnosis and Treatment of 2 Sepsis”, Jingqian Su et al., based on the published literatures and related biomarkers, explored the potential of DcR3 in the diagnosis and treatment of sepsis.
The comments and suggestions for this manuscript are as follows:
1. The introduction, and the Summary and Outlook of the manuscript is a typical textbook type. This is lacking intellectual input from authors. The author must provide a comprehensive introduction and summary with proper references.
2. The sections of the main text are lacking a proper connection and flow of texts. For example, in section 3 “sepsis” the author describes the first disease etiology and pathophysiology of sepsis (1 and 2 paragraph), while paragraph 3 discusses about origin of the terminology. The author must rearrange the text paragraph for better understanding.
3. Page 10 line 311-313. The statement “DcR3 transgenic mice had reduced levels of pro-inflammatory cytokines and chemokines in bronchoalveolar lavage fluid and decreased lung infiltration” and page 11 line 324-327, the statement “However, because the DcR3 gene is not present in the rat and mouse genomes, it is impossible to construct DcR3 knockout mice, making the application of DcR3 in murine models somewhat limited and presenting a unique challenge to preclinical translational studies on DcR3”. Both statements are contradictory. The author must take care of this issue and revise it carefully.
4. Figure 1-5. The graphical presentations of figures are up to mark and well-illustrated. The author must provide the detailed figure legends for each figure.
Minor editing of the English language required
Author Response
August 10, 2023
Prof. Dr. Maurizio Battino
Editor-in-Chief
Ms. Beiner Zhang
Assistant Editor
International Journal of Molecular Sciences
Dear Editor,
I wish to re-submit the manuscript titled “Research Progress of DcR3 in the Diagnosis and Treatment of Sepsis.” The manuscript ID is ijms-2505432.
We express our gratitude to you and the reviewers for your valuable suggestions and insightful perspectives. The manuscript has greatly benefited from these astute recommendations.
Attached is the revised version of our manuscript. In the following pages are our point-by-point responses to each of the comments of the reviewers. Revisions in the text are highlighted by the utilization of the color red. We hope that the revisions in the manuscript and our accompanying responses would be sufficient to make our manuscript suitable for publication in International Journal of Molecular Sciences.
Thank you for your consideration. I look forward to hearing from you.
Sincerely,
Jingqian Su, Ph.D.
Associate Professor
Fujian Key Laboratory of Innate Immune Biology
Biomedical Research Center of South China
College of Life Science, Fujian Normal University
Fuzhou 350117, Fujian, China
Tel: +86-18950498937
E-mail: sjq027@fjnu.edu.cn
Responses to the comments of Reviewer #2
1) The introduction, and the Summary and Outlook of the manuscript is a typical textbook type. This is lacking intellectual input from authors. The author must provide a comprehensive introduction and summary with proper references.
Response:
We would like to extend our heartfelt appreciation to the reviewer for their invaluable contributions in augmenting the caliber of the article. The comprehensive introduction and summary with proper references have added in the he abstracts, introduction, discussion and outlook content (Lines 15-16, 20-21, 41-46, 354-355, 356-357, and 389-390), the details are as follows:
(1) Lines 15-16
DcR3 alone or in combination with other markers has shown promising results in the early diagnosis of sepsis.
(2) Lines 20-21
The application of DcR3 in the treatment of a mouse model of sepsis also achieved good efficacy.
(3) Lines 41-46
DcR3 alone or in combination with other markers has shown promising results in the early diagnosis of sepsis, due to the multiple functions of DcR3 in immunomodulation. Previously, DongYu Liang et al. (2015) found that in vitro injection of DcR3-Fc (Decoy receptor 3 fused to Fc fragment of IgG1) recombinant protein increased survival, decreased inflammation levels, and improved organ damage in septic mice [9].
(4) Lines 354-355
Animal experiments have confirmed that the use of DcR3 for sepsis treatment is feasible and effective [8].
(5) Lines 356-357
Future research on DcR3 should focus on elaborating the mechanism of action of DcR3 in ameliorating sepsis, as well as promoting the clinical application of DcR3.
(6) Lines 389-390
This is the reason for the lack of effective drugs for sepsis treatment on the market.
2) The sections of the main text are lacking a proper connection and flow of texts. For example, in section 3 “sepsis” the author describes the first disease etiology and pathophysiology of sepsis (1 and 2 paragraph), while paragraph 3 discusses about origin of the terminology. The author must rearrange the text paragraph for better understanding.
Response:
First, thank you very much for your valuable comments. After your reminder, Upon reevaluating the hierarchical arrangement of the article, we have identified certain inconsistencies within its structure. In light of the recommendations provided, we have undertaken a reorganization of the exposition on "sepsis" to commence with a comprehensive definition of sepsis and its associated perils. Subsequently, we have incorporated a discourse on the etymology of sepsis terminology and the evolution of its definitions, followed by an elucidation of sepsis pathophysiology. Lastly, we have included a thorough examination of early diagnosis, scoring tools, and the treatment of sepsis. The specific modifications are as follows:
(1) Lines 153-166
The word sepsis is derived from the Greek word σήψη, meaning putrefaction. The term has been used since the time of Hippocrates, but the definition has been ambiguous in the medical community, leading to numerous difficulties in clinical diagnosis and medication prescription [56,59]. To overcome this inconsistency, the definition of sepsis has been continuously revised in recent decades. The original definition of sepsis (sepsis 1.0) was developed at the consensus meeting of the American College of Chest Physicians/Society of Critical Care Medicine in 1991, subsequently updated in 2001 (sepsis 2.0), and the most recent definition in use today is sepsis 3.0, published in 2016 [5,60,61]. Sepsis 3.0 is defined as a life-threatening organ dysfunction caused by the dysregulation of the body's response to infection, and septic shock is the most severe manifestation of sepsis with severe circulatory, cellular, and metabolic dysfunction and a greater risk of death [5,62]. Sepsis 3.0 emphasizes that sepsis is caused by infection and can be distinguished from non-infectious causes of organ dysfunction and common infections that do not cause organ dysfunction.
(2) Lines 181-184
According to sepsis 3.0, it is a disease caused by an overreaction of the host to infection. The occurrence of sepsis originates from various infections, including pathogenic microbial infections and various traumas, the former being more common in clinical practice.
3) Page 10 line 311-313. The statement “DcR3 transgenic mice had reduced levels of pro-inflammatory cytokines and chemokines in bronchoalveolar lavage fluid and decreased lung infiltration” and page 11 line 324-327, the statement “However, because the DcR3 gene is not present in the rat and mouse genomes, it is impossible to construct DcR3 knockout mice, making the application of DcR3 in murine models somewhat limited and presenting a unique challenge to preclinical translational studies on DcR3”. Both statements are contradictory. The author must take care of this issue and revise it carefully.
Response:
Thank you very much for being able to clearly point out the problems with the article. Here are some of my insights into the problem: DcR3, a pleiotropic immunomodulator, is found in various species, including humans. However, it is not absent in mice (Mus musculus) and rats (Rattus norvegicus). Consequently, investigating the role of DcR3 in the mouse sepsis model presents a challenge due to the inability to directly create DcR3 knockout mice. Fortunately, there are alternative options available, such as the presence of FasL, LIGHT, and TL1A in mice, which can bind to human DcR3. By constructing DcR3 transgenic mice, researchers can explore the involvement of DcR3 in mouse sepsis models. Based on reviewer’s suggestions, we have made some changes to the previous statement (Lines 357-363) , the details are as follows:
(1) Lines 357-363
Since the DcR3 gene does not exist in the rat and mouse genome, it is not possible to directly construct DcR3 knockout mice. Although the therapeutic effects of DcR3 in septic mice can be investigated by constructing DcR3-expressing mice or by administering DcR3 in vitro, this is a more demanding experimental technique for the operator and is both time-consuming and expensive, making the application of DcR3 in the murine model somewhat limited.
4) Figure 1-5. The graphical presentations of figures are up to mark and well-illustrated. The author must provide the detailed figure legends for each figure.
Response:
We express our sincere gratitude to the editors for their invaluable contributions in enhancing the quality of the article. The detailed figure legends have been added in figures (Lines 76-83, 93-98, 201-206, 225-226, and 329-334), the details are as follows:
(1) Lines 76-83
LPS: Lipopolysaccharide; PG: Peptidoglycan; LTA: Lipoteichoic acid; TLR4: Toll-like receptor 4; TLR2: Toll-like receptor 2; TRIF: Tir domain-containing adaptor inducing interferon-beta; MYD88: Myeloid differentiation factor 88; TAK1: Transforming growth factor-beta-activated kinase 1; TAB1/2: Transforming growth factor beta-activated kinase 1 binding protein 1/2; IKKβ: IkappaB kinase β; IKKα: IkappaB kinase α; NEMO: Nuclear factor kappaB essential modulator; PI3K: Phosphoinositide 3 kinase ; PDK1: 3-phosphoinositide-dependent protein kinase 1; AKT: Protein kinase B; NF-κB: Nuclear factor kappaB; DcR3: Decoy receptor 3; P50: Nuclear factor kappaB1; P65: Nuclear factor kappa B2.
(2) Lines 93-98
LIGHT: Tumor necrosis factor ligand superfamily member 14; TL1A: TNF-like molecule 1A; FasL: Fas ligand; HVEM: Herpesvirus Entry Mediator; LTβR: Lymphotoxin-beta receptor; DcR3: Decoy receptor 3; DR3: Death receptor 3; Fas: TNF superfamily receptor 6; CD44v3: Cluster of differentiation 44 variant 3; ICAM-1: Intercellular adhesion molecule 1; VCAM-1: Vascular cell adhesion molecule-1; DC: Dendritic cell; T: T lymphocytes; Th2: T-helper 2 cells.
(3) Lines 201-206
DAMPs: Damage-associated molecular patterns; PAMPs: Pathogen-associated molecular patterns; PRRs: Pattern recognition receptors; HMGB1: High mobility group box-1 protein; C3a: Complement component 3a; C5a: Complement component 5a; IL-1β: Interleukin-1beta; IL-6: Interleukin-6; TNF-α: Necrosis Factor alpha; NO: Nitrous oxide; IL-4: Interleukin-4; IL-10: Interleukin-10; IL-37: Interleukin-37; TGF-β: Transforming growth factor-beta; Treg: Regulatory T-cells.
(4) Lines 225-226
PCT: Procalcitonin; CRP: C-reactive protein; IL-6: Interleukin-6; IL-10: Interleukin-4; HLA-DR: Human leukocyte antigen DR; CD64: Cluster of differentiation 64.
(5) Lines 329-334
M0: M0-like macrophage; M1: M1-like macrophage; M2: M2-like macrophage; CD4+T: CD4 positive lymphocytes; Th1: T-helper 1 cells; Th2: T-helper 2 cells; IL-1β: Interleukin-1beta; IL-6: Interleukin-6; TNF-α: Necrosis Factor alpha; NO: Nitrous oxide; DcR3: Decoy receptor 3; Fas: TNF superfamily receptor 6; FasL: Fas ligand; LIGHT: Tumor necrosis factor ligand superfamily member 14; LTβR: Lymphotoxin-beta receptor; TL1A: TNF-like molecule 1A; DR3: Death receptor 3.
5) Minor editing of the English language required
Response:
We would like to express our profound appreciation to the editors for their invaluable suggestions. The manuscript underwent editing by Editage, a professional language editing company.

Reviewer 3 Report
In this work the Authors discuss the reliability of Decoy receptor 3 (DcR3) as a biomarker of sepsis. Furthermore, they highlighted its possible implications in the treatment of this insidious clinical entity. The concept of finding always new biomarkers/treatment able to help physicians in facing sepsis is really stimulating.
Overall, this manuscript is well-designed and provides different interesting aspects. Anyway, there are different concerns that should be addressed:
1. Lines 12-13: the Authors stated “currently, there are no specific drugs available for treatment”. This sentence might be misread. Please try to clarify it.
2. Please avoid useless abbreviation in the abstract (e.g. TNFSF6, TNFSF14 and TNFSF15). If the Authors need these abbreviation in the main text, they should report them in the appropriate parts.
3. Lines 34-36: the introduction to the concept and main problems in sepsis are well-described- However, I can’t ignore that the references did not include two important work: 1) the most recent guidelines by Evans et al. (Evans L, Rhodes A, Alhazzani W, Antonelli M, Coopersmith CM, French C, Machado FR, Mcintyre L, Ostermann M, Prescott HC, Schorr C, Simpson S, Wiersinga WJ, Alshamsi F, Angus DC, Arabi Y, Azevedo L, Beale R, Beilman G, Belley-Cote E, Burry L, Cecconi M, Centofanti J, Coz Yataco A, De Waele J, Dellinger RP, Doi K, Du B, Estenssoro E, Ferrer R, Gomersall C, Hodgson C, Hylander Møller M, Iwashyna T, Jacob S, Kleinpell R, Klompas M, Koh Y, Kumar A, Kwizera A, Lobo S, Masur H, McGloughlin S, Mehta S, Mehta Y, Mer M, Nunnally M, Oczkowski S, Osborn T, Papathanassoglou E, Perner A, Puskarich M, Roberts J, Schweickert W, Seckel M, Sevransky J, Sprung CL, Welte T, Zimmerman J, Levy M. Surviving Sepsis Campaign: International Guidelines for Management of Sepsis and Septic Shock 2021. Crit Care Med. 2021 Nov 1;49(11):e1063-e1143); and 2) one of the most recent manuscripts on sepsis treatment by Guarino et al. (Guarino M, Perna B, Cesaro AE, Maritati M, Spampinato MD, Contini C, De Giorgio R. 2023 Update on Sepsis and Septic Shock in Adult Patients: Management in the Emergency Department. J Clin Med. 2023 Apr 28;12(9):3188). Please include these two references in the manuscript.
4. Many abbreviations are lacking (e.g. lines 40, 47, 67, 73, 79). Please fix it.
5. Figures presented are really well designed and help the reader to understand all the biochemical process described. However, I rather prefer that the figure legends stay above the figures and that all of them were followed by notes highlighting the abbreviation included in each image.
6. Lines 126-127: the Authors stated that “there is currently a lack of specific pharmaceutical interventions for its 126 treatment”. Actually, I strongly disagree with this sentence, addressing the Authors to the two references suggested above (that, once again, I suggest to add even in this case). Sepsis has a multifaced treatment that has the bilateral goal of facing the infection and support the cardiorespiratory function of the patient. I imagine that the Authors meant the lack of a specific treatment able to immediately interrupt the cytokine cascade and the subsequent microcirculatory damage induced by the dysregulated host response. In this case, the sentence should be rephrased in order to avoid misreading.
7. Table 1: this table is quite interesting. However, why did the Authors not include presepsin and pro-adrenomedullin? These two markers are really promising and, to date, quite diffuse in sepsis detection.
8. Line 250: the Authors stated that “Blood culture is the gold standard for sepsis detection”. Even in this case I strongly disagree with the Authors. Saying so, you lead a reader to think that if blood cultures are negative the patient is not septic. This is absolutely incorrect! Furthermore, what about those non-bacteriemic infections (e.g. abscess)? Do they not cause sepsis? Rephrase this part and clarify this concept.
Minor English revisions are advisable
Author Response
August 10, 2023
Prof. Dr. Maurizio Battino
Editor-in-Chief
Ms. Beiner Zhang
Assistant Editor
International Journal of Molecular Sciences
Dear Editor,
I wish to re-submit the manuscript titled “Research Progress of DcR3 in the Diagnosis and Treatment of Sepsis.” The manuscript ID is ijms-2505432.
We express our gratitude to you and the reviewers for your valuable suggestions and insightful perspectives. The manuscript has greatly benefited from these astute recommendations.
Attached is the revised version of our manuscript. In the following pages are our point-by-point responses to each of the comments of the reviewers. Revisions in the text are highlighted by the utilization of the color red. We hope that the revisions in the manuscript and our accompanying responses would be sufficient to make our manuscript suitable for publication in International Journal of Molecular Sciences.
Thank you for your consideration. I look forward to hearing from you.
Sincerely,
Jingqian Su, Ph.D.
Associate Professor
Fujian Key Laboratory of Innate Immune Biology
Biomedical Research Center of South China
College of Life Science, Fujian Normal University
Fuzhou 350117, Fujian, China
Tel: +86-18950498937
E-mail: sjq027@fjnu.edu.cn
Responses to the comments of Reviewer #3
1) Lines 12-13: the Authors stated “currently, there are no specific drugs available for treatment”. This sentence might be misread. Please try to clarify it.
Response:
Thank you very much for your valuable comments. We apologize for the misunderstanding caused by this statement, and have revised this statement, as well as several similar statements in the text. The changes are as follows:
(1) Lines 11-13
Currently, no specific drug that can alleviate or even cure sepsis in a comprehensive and multi-level manner has been found.
(2) Lines 142-144
No specific drug has yet been found that can alleviate or even cure sepsis in a comprehensive and multi-level manner [52].
2) Please avoid useless abbreviation in the abstract (e.g., TNFSF6, TNFSF14 and TNFSF15). If the Authors need this abbreviation in the main text, they should report them in the appropriate parts.
Response:
We express our sincere gratitude for the invaluable guidance provided regarding the art of writing. As per your astute observation, certain abbreviations found in the abstract lacked coherence and have been duly eliminated, as per your recommendation. Furthermore, in light of your timely reminder, we have meticulously reassessed the presence of other abbreviations within the text, scrutinizing their necessity, resulting in the subsequent modifications outlined below:
(1) Lines 27-30
DcR3 can competitively bind FasL (TNF superfamily receptor 6, TNFSF6), LIGHT (Tumor necrosis factor ligand superfamily member 14, TNFSF14), and TL1A (TNF-like molecule 1A) with a strong affinity through decoy action and block downstream signaling;
3) Lines 34-36: the introduction to the concept and main problems in sepsis are well-described- However, I can’t ignore that the references did not include two important work: 1) the most recent guidelines by Evans et al. (Evans L, Rhodes A, Alhazzani W, Antonelli M, Coopersmith CM, French C, Machado FR, Mcintyre L, Ostermann M, Prescott HC, Schorr C, Simpson S, Wiersinga WJ, Alshamsi F, Angus DC, Arabi Y, Azevedo L, Beale R, Beilman G, Belley-Cote E, Burry L, Cecconi M, Centofanti J, Coz Yataco A, De Waele J, Dellinger RP, Doi K, Du B, Estenssoro E, Ferrer R, Gomersall C, Hodgson C, Hylander Møller M, Iwashyna T, Jacob S, Kleinpell R, Klompas M, Koh Y, Kumar A, Kwizera A, Lobo S, Masur H, McGloughlin S, Mehta S, Mehta Y, Mer M, Nunnally M, Oczkowski S, Osborn T, Papathanassoglou E, Perner A, Puskarich M, Roberts J, Schweickert W, Seckel M, Sevransky J, Sprung CL, Welte T, Zimmerman J, Levy M. Surviving Sepsis Campaign: International Guidelines for Management of Sepsis and Septic Shock 2021. Crit Care Med. 2021 Nov 1;49(11):e1063-e1143); and 2) one of the most recent manuscripts on sepsis treatment by Guarino et al. (Guarino M, Perna B, Cesaro AE, Maritati M, Spampinato MD, Contini C, De Giorgio R. 2023 Update on Sepsis and Septic Shock in Adult Patients: Management in the Emergency Department. J Clin Med. 2023 Apr 28;12(9):3188). Please include these two references in the manuscript.
Response:
Thank you very much for your careful review of the references in this paper. The work done by Evans et al. and Guarino et al. that you mentioned is indeed very important and both articles help us to understand sepsis care and related treatment strategies. The two references have been incorporated into this paper under the corresponding numbers [69] and [87] (Line 208 and 251). The specific modifications are as follows:
(1) Lines 590-597
- Evans, L.; Rhodes, A.; Alhazzani, W.; Antonelli, M.; Coopersmith, C. M.; French, C.; Machado, F. R.; McIntyre, L.; Ostermann, M.; Prescott, H. C.; Schorr, C.; Simpson, S.; Wiersinga, W. J.; Alshamsi, F.; Angus, D. C.; Arabi, Y.; Azevedo, L.; Beale, R.; Beilman, G.; Belley-Cote, E.; Burry, L.; Cecconi, M.; Centofanti, J.; Coz Yataco, A.; De Waele, J.; Dellinger, R. P.; Doi, K.; Du, B.; Estenssoro, E.; Ferrer, R.; Gomersall, C.; Hodgson, C.; Hylander Møller, M.; Iwashyna, T.; Jacob, S.; Kleinpell, R.; Klompas, M.; Koh, Y.; Kumar, A.; Kwizera, A.; Lobo, S.; Masur, H.; McGloughlin, S.; Mehta, S.; Mehta, Y.; Mer, M.; Nunnally, M.; Oczkowski, S.; Osborn, T.; Papathanassoglou, E.; Perner, A.; Puskarich, M.; Roberts, J.; Schweickert, W.; Seckel, M.; Sevransky, J.; Sprung, C. L.; Welte, T.; Zimmerman, J.; Levy, M., Surviving Sepsis Campaign: International Guidelines for Management of Sepsis and Septic Shock 2021. Crit. Care. Med. 2021, 49 (11), e1063-e1143. DOI: 10.1097/CCM.0000000000005337
(2) Lines 635-636
- Guarino, M.; Perna, B.; Cesaro, A. E.; Maritati, M.; Spampinato, M. D.; Contini, C.; De Giorgio, R., 2023 Update on Sepsis and Septic Shock in Adult Patients: Management in the Emergency Department. J. Clin. Med. 2023, 12 (9). DOI: 10.3390/jcm12093188
4) Many abbreviations are lacking (e.g., lines 40, 47, 67, 73, 79). Please fix it.
Response:
We express our sincere gratitude to the editors for their invaluable contributions in enhancing the quality of the article. The abbreviations have been added in text, the details are as follows:
(1) Line 44
DcR3-Fc (Decoy receptor 3 fused to Fc fragment of IgG1)
(2) Lines 51-52
TNFRSF6B (tumor necrosis factor receptor superfamily member 6b)
(3) Lines 72-73
nuclear factor kappaB (NF-κB)ï¼›phosphoinositide 3 kinase-protein kinase B (PI3K-AKT)
(4) Line 91
T-helper 1/2 cells (Th1/Th2)
(5) Line 268
interferon γ (IFN-γ)
5) Figures presented are well designed and help the reader to understand all the biochemical process described. However, I rather prefer that the figure legends stay above the figures and that all of them were followed by notes highlighting the abbreviation included in each image.
Response:
We express our sincere gratitude to the reviewer for the invaluable contributions in enhancing the quality of the article. The abbreviations have been added in figure legends, the details are as follows:
(1) Lines 76-83
LPS: Lipopolysaccharide; PG: Peptidoglycan; LTA: Lipoteichoic acid; TLR4: Toll-like receptor 4; TLR2: Toll-like receptor 2; TRIF: Tir domain-containing adaptor inducing interferon-beta; MYD88: Myeloid differentiation factor 88; TAK1: Transforming growth factor-beta-activated kinase 1; TAB1/2: Transforming growth factor beta-activated kinase 1 binding protein 1/2; IKKβ: IkappaB kinase β; IKKα: IkappaB kinase α; NEMO: Nuclear factor kappaB essential modulator; PI3K: Phosphoinositide 3 kinase ; PDK1: 3-phosphoinositide-dependent protein kinase 1; AKT: Protein kinase B; NF-κB: Nuclear factor kappaB; DcR3: Decoy receptor 3; P50: Nuclear factor kappaB1; P65: Nuclear factor kappa B2.
(2) Lines 93-98
LIGHT: Tumor necrosis factor ligand superfamily member 14; TL1A: TNF-like molecule 1A; FasL: Fas ligand; HVEM: Herpesvirus Entry Mediator; LTβR: Lymphotoxin-beta receptor; DcR3: Decoy receptor 3; DR3: Death receptor 3; Fas: TNF superfamily receptor 6; CD44v3: Cluster of differentiation 44 variant 3; ICAM-1: Intercellular adhesion molecule 1; VCAM-1: Vascular cell adhesion molecule-1; DC: Dendritic cell; T: T lymphocytes; Th2: T-helper 2 cells.
(3) Lines 201-206
DAMPs: Damage-associated molecular patterns; PAMPs: Pathogen-associated molecular patterns; PRRs: Pattern recognition receptors; HMGB1: High mobility group box-1 protein; C3a: Complement component 3a; C5a: Complement component 5a; IL-1β: Interleukin-1beta; IL-6: Interleukin-6; TNF-α: Necrosis Factor alpha; NO: Nitrous oxide; IL-4: Interleukin-4; IL-10: Interleukin-10; IL-37: Interleukin-37; TGF-β: Transforming growth factor-beta; Treg: Regulatory T-cells.
(4) Lines 225-226
PCT: Procalcitonin; CRP: C-reactive protein; IL-6: Interleukin-6; IL-10: Interleukin-4; HLA-DR: Human leukocyte antigen DR; CD64: Cluster of differentiation 64.
(5) Lines 329-334
M0: M0-like macrophage; M1: M1-like macrophage; M2: M2-like macrophage; CD4+T: CD4 positive lymphocytes; Th1: T-helper 1 cells; Th2: T-helper 2 cells; IL-1β: Interleukin-1beta; IL-6: Interleukin-6; TNF-α: Necrosis Factor alpha; NO: Nitrous oxide; DcR3: Decoy receptor 3; Fas: TNF superfamily receptor 6; FasL: Fas ligand; LIGHT: Tumor necrosis factor ligand superfamily member 14; LTβR: Lymphotoxin-beta receptor; TL1A: TNF-like molecule 1A; DR3: Death receptor 3.
6) Lines 126-127: the Authors stated that “there is currently a lack of specific pharmaceutical interventions for its 126 treatment”. Actually, I strongly disagree with this sentence, addressing the Authors to the two references suggested above (that, once again, I suggest to add even in this case). Sepsis has a multifaced treatment that has the bilateral goal of facing the infection and support the cardiorespiratory function of the patient. I imagine that the Authors meant the lack of a specific treatment able to immediately interrupt the cytokine cascade and the subsequent microcirculatory damage induced by the dysregulated host response. In this case, the sentence should be rephrased in order to avoid misreading.
Response:
Thank you for your patient guidance on the issues expressed in this article. The aforementioned statement, "there is a lack of specific drugs to treat the disease," aims to highlight the absence of medications capable of simultaneously alleviating excessive inflammation and enhancing immunosuppression. I regret any potential confusion caused by a prior assertion, as numerous drugs have indeed been formulated for sepsis treatment, including antimicrobials and fluid resuscitation. However, it is important to note that these drugs do not address sepsis from multiple perspectives. The previous formulation has been modified in accordance with your suggestions, the details are as follows:
(1) Lines 11-13
Currently, no specific drug that can alleviate or even cure sepsis in a comprehensive and multi-level manner has been found.
(2) Lines 142-144
No specific drug has yet been found that can alleviate or even cure sepsis in a comprehensive and multi-level manner [52].
7) Table 1: this table is quite interesting. However, why did the Authors not include presepsin and pro-adrenomedullin? These two markers are really promising and, to date, quite diffuse in sepsis detection.
Response:
We would like to express our profound appreciation to the editors for their invaluable suggestions. In recent years, sepsis-related biomarkers have been discovered, and among the many members of sepsis biomarkers, presepsin and pro-adrenomedullin are undoubtedly very promising. Presepsin and pro-adrenomedullin are significantly elevated during sepsis, and presepsin can be used to some extent to determine bacterial infection. Since this article consists of a place dedicated to sepsis biomarkers (Table 1), presepsin and pro-adrenomedullin have been added and categorized as hormone-related based on their properties. It was also noted in the Discussion and Outlook that DcR3 could be considered for sepsis prediction in the future in conjunction with presepsin and pro-adrenomedullin. In addition, to have a clearer understanding of the role of presepsin and pro-adrenomedullin in sepsis biomarkers, two new references were cited, numbered [76] and [77]. The specific modifications are as follows:
(1) Lines 217-220
As shown in Table 1, sepsis biomarkers include acute phase response proteins, cytokines and chemokines, cell surface and soluble receptors, vascular endothelium-related markers, coagulation-related markers, neurotransmitter-related markers, and hormone-related markers [72-77].
(2) Line 224
Table 1. Relevant biomarkers of sepsis.
Function |
Markers |
Acute phase reactive protein |
PCT; CRP |
Cytokines and chemokines |
IL-6; IL-10 |
Cell surface and soluble receptors |
HLA-DR; CD64 |
Vascular endothelium-related |
cell adhesion molecules; angiopoietin |
Coagulation-related |
antithrombin III |
Neurotransmitter-related |
butyrylcholinesterase |
Hormone-related |
presepsin; pro-adrenomedullin |
(3) Lines 371-373
DcR3 can be used as a biomarker for sepsis, and can be used alone for early diagnosis or in combination with markers such as CRP, PCT, butyrylcholinesterase, presepsin, and pro-adrenomedullin.
(4) Lines 614-617
- Velissaris, D.; Zareifopoulos, N.; Karamouzos, V.; Karanikolas, E.; Pierrakos, C.; Koniari, I.; Karanikolas, M., Presepsin as a Diagnostic and Prognostic Biomarker in Sepsis. Cureus 2021, 13 (5), e15019. DOI: 10.7759/cureus.15019
- Önal, U.; Valenzuela-Sánchez, F.; Vandana, K. E.; Rello, J., Mid-Regional Pro-Adrenomedullin (MR-proADM) as a Biomarker for Sepsis and Septic Shock: Narrative Review. Healthcare (Basel) 2018, 6 (3). DOI: 10.3390/healthcare6030110
8) Line 250: the Authors stated that “Blood culture is the gold standard for sepsis detection”. Even in this case I strongly disagree with the Authors. Saying so, you lead a reader to think that if blood cultures are negative the patient is not septic. This is absolutely incorrect! Furthermore, what about those non-bacteriemic infections (e.g., abscess)? Do they not cause sepsis? Rephrase this part and clarify this concept.
Response:
We would like to extend our heartfelt appreciation to the reviewer for the invaluable contributions in augmenting the caliber of the article. The relevant statement has been changed according to your suggestion, and the details of the change are as follows:
(1) Lines 213-216
However, these are time-consuming and prone to false-negative results, and cannot be used to diagnose non-bacterial causes of sepsis, which is not conducive to the early diagnosis of sepsis and the timely intervention of antibiotics [71].
(2) Lines 274-279
Blood culture is the gold standard for sepsis detection, but this test is time consuming (48 h-72 h), has a low positive rate and an inability to diagnose sepsis caused by non-bacterial infections, therefore, there is an urgent need to find some efficient and sensitive testing methods for use in clinics [94]. The discovery of specific biomarkers has alleviated this situation, with early diagnosis of sepsis relying on specific biomarkers, but these also have certain limitations [78].
9) Comments on the Quality of English Language-Minor English revisions are advisable
Response:
We would like to express our profound appreciation to the editors for their invaluable suggestions. The manuscript underwent editing by Editage, a professional language editing company.

Round 2
Reviewer 1 Report
Excellent work. All requested changes were addressed accordingly. It can be accepted for publication without further revision.
Reviewer 3 Report
I would like to thank the Authors for all the work done. Almost all my concerns hav been amended. The manuscript is now suitable for publication.